# Diagnostic and Treatment Challenges of Emergent COVID-Associated-Mucormycosis: A Case Report and Review of the Literature

**DOI:** 10.3390/antibiotics12010031

**Published:** 2022-12-25

**Authors:** Manuela Arbune, Anca-Adriana Arbune, Alexandru Nechifor, Iulia Chiscop, Violeta Sapira

**Affiliations:** 1Clinical Medical Department, “Dunarea de Jos” University from Galati, 800008 Galati, Romania; 2Neurology Clinic, Fundeni Clinical Institute, 022328 Bucharest, Romania; 3Clinical Surgical Department, “Dunarea de Jos” University from Galati, 800008 Galati, Romania

**Keywords:** COVID-19, mucormycosis, rhino-orbital mucormycosis, superinfection, healthcare associated infection

## Abstract

Mucormycosis is a rare fungal infection, with high mortality, commonly associated with diabetes, malignancies, immunosuppressive therapy, and other immunodeficiency conditions. The emergence of mucormycosis cases has been advanced by the COVID-19 pandemic. Clinical presentation is variable, from asymptomatic to persistent fever or localized infections. We present a case of a Romanian old man, without diabetes or other immunodepression, with COVID-19 who developed severe rhino-orbital mucormycosis and bacterial superinfections, with Pseudomonas aeruginosa and Klebsiella pneumoniae. The late diagnostic and antifungal treatment was related to extensive lesions, bone and tissue loss, and required complex reconstruction procedures. We review the relationships between mucormycosis, COVID-19, and bacterial associated infections. The suspicion index of mucormycosis should be increased in medical practice. The diagnostic and treatment of COVID-19-Associated-Mucormycosis is currently challenging, calling for multidisciplinary collaboration.

## 1. Introduction

The first case of a mucormycosis was reported in 1876, and was related to a German patient who died of cancer [1]. The present incidence and prevalence of mucormycosis are not well-known, due to the difficulty in acquiring deep tissue samples, the low sensitivity of the diagnosis tests, and the incoherence of public health notification of infection around the world [1]. Mucormycosis was registered more frequently in India, but the infection associated with immunosuppression was also reported in other countries in Europe and the Americas [2].

The order Mucorales includes genera of filamentous fungi that can cause human infections. These microorganisms are rapidly grown, releasing a considerable number of spores that are spread by air in the environment. Mucorales fungi are mostly found in thrush, decaying vegetation, dust, or soil [3]. Human exposure to these fungi occurs during daily activities, and infections could be transmitted by inhaling spores, ingesting contaminated food, or inoculating the damaged tissues of skin and mucous membranes.

Genera *Rhizopus*, *Mucor* and Rhizomucor are more frequently attributable to human infections. *Cunninghamella*, *Lichtheimia*, *Saksenaea*, and *Apophysomyces* are less frequently identified genera [4].

The presumptive identification of mucorales hyphae from biological products is based on microscopic characterization; it has a thick appearance, a diameter between 5 and 15 microns, irregular ramifications, and few septa. These features are distinctive from ascomycetes, which have the appearance of narrower hyphae, with regular ramifications, and multiple septa [1,4]. The positive results must be considered in the clinical context, judging the possible colonization of the airways or contamination of cultures, which are different from infection. The serological reaction of 1,3-beta-D-glucan could be alternative diagnosis. Molecular tests based on polymerase chain reaction and matrix-assisted laser desorption ionization-time of flight (MALDI-TOF) sequencing techniques allow the identification of species from fungal cultures [5].

Mucormycosis is a rare infection in immunocompetent persons. Most cases are opportunistic infections, associated with immunosuppressive conditions, such as diabetes, malignant diseases, oncological chemotherapy/immunotherapy, transplantation of organs or hematopoietic stem cells, severe neutropenia, graft-versus-host rejection, hypersideremia, acquired immunodeficiency syndrome, use of injectable drugs, trauma or burns, and malnutrition [6].

The vascular invasion of the hyphae, consecutive tissue infarction and necrosis, are the pathogenic stages of mucorales infection.

The clinical appearance of mucormycosis could be systemic infections or localized infections to the anatomical regions as rhino-orbito-sinus, cerebral, lung, gastro-intestinal tract, kidney, cutaneous, or mucosal membranes [7].

Added to the clinical diagnosis, the imaging investigations are required to specify the extension of the lesions. Other investigations, such as broncho-alveolar lavage, esophageal, gastric, or intestinal endoscopy, are useful, depending on the anatomical site of the fungal infection. Diagnosis is confirmed by biopsy, with anatomopathological examination of biological samples by Hematoxylin–Eosin, Periodic Acid Schiff (PAS), or Grocott–Gomori staining and supplementary immunohistochemical examination. Fungal culture is the gold standard for confirming the diagnosis, as it is advantageous for susceptibility testing to antifungal drugs. However, the sensitivity of mucorales cultures is low, with 50% false negative results [1].

The recommended therapeutic strategy for mucormycosis is to combine the surgical removal of the damaged tissues and antifungal medication. The first line drug is intravenous liposomal amphotericin. Posaconazole or Isavuconazole could be used to de-escalate or as a salvage prescription in unresponsive or intolerant patients to conventional treatment [1,5].

Mortality ranges from 40% to 80%, depending on either the location and expanse of the lesions, or the severity of the host’s immunodepression [2,5,6].

The real burden of human mucormycosis is unknown, because is not a reportable disease [8]. A review of 851 cases collected from January 2000 to January 2017 evidenced that the majority of the cases were from Europe (34%), followed by Asia (31%), and North or South America (28%) [9]. The disease is more frequent in developing countries; however, the diagnosis is underestimated in this region, mainly due to laboratories with sub-optimal facilities [8].

A report from 2016 estimated an annual incidence of mucormycosis in Romania of 0.04/100,000 [10]. The global prevalence of mucormycosis estimates of the year 2019–2020 ranges between 0.005 and 1.7 per million population, where the highest prevalence is noticed in India (0.14 per 1000) [11]. The growing number of mucormycosis reported cases during the COVID-19 pandemic increased the interest of the public health sector.

## 2. Case Report

A 69-year-old Caucasian male patient, from an urban area, was presented to the Infectious Diseases Daily Clinic for severe stomatitis in the third month after he was hospitalized for severe COVID-19 in the intensive care unit, during the fourth pandemic wave. The past medical history was irrelevant, excepting the blood hypertension and the use of a removable prosthetic denture. He was not vaccinated for SARS-CoV-2 and had developed complications with bronchopneumonia and respiratory failure. The markers for HIV, and hepatitis B and C were negative. During COVID-19 hospitalization, the computer tomographic exam found 75% lung lesions, with bilateral “ground glass” opacities. He was treated according to the local protocols, with Favipiravir (Remdesivir was temporarily unavailable), corticosteroid (Dexamethasone), anticoagulant (Fraxiparine), antibiotics (Ceftriaxone and additional Ciprofloxacine), and oxygen support with continuous positive airway pressure (CPAP). The response of biological markers was unfavorable, regarding the increase of C-Reactive Protein (CRP), D-dimers, and leucocyte index (N/Ly) during the hospital course. He was discharged after 14 days, when the respiratory failure was improved, but he returned home being dependent on a mobile device for oxygen supply. He was very asthenic and had difficulty eating, due to extensive stomatitis including the gums and the palate (Table A1; Figure 1). He received a 2-week course of Fluconazole, with no clinical response.

Moreover, in the third day after discharge (Day-17), he had diarrhea with *Clostridium difficile* (GDH positive, Toxine A and B positive), a health care infection with favorable evolution after 10 days of ambulatory treatment with oral rehydration and oral Vancomycin (125 mg/day every 6 h).

On the 14th day after discharge (Day-28), the left hemiparesis “explosion” of the left eyeball and unilateral visual loss and had appeared upon waking-up in the morning. The ophthalmological examination of the left eye described palpebral ptosis, ocular motility limitation in all directions, palpebral oedema, axial exophthalmos, unreactive mydriasis, chemosis, and subconjunctival hemorrhage in all quadrants (Figure 2).

The ophthalmoscopy found a blurred outline on the papilla of the optic nerve, but it was normally colored. There were dilated retinal vessels with tortuous paths at the central emergence, and dot-blot hemorrhages in the retina’s upper quadrants, although there were no apparent macular lesions. The ambulatory diagnostic was a post-COVID stroke, and a brain imaging examination was scheduled, but the patient decided to be cared for at home.

The bacterial cultures of the oral lesions were positive for *Klebsiella pneumoniae* and *Pseudomonas aeruginosa*. According to the antibiotic sensitivity (Vitek2), a 10-day course of Cefuroxime and Gentamicine was ordered, but persistent positive bacterial cultures and positive fungal cultures with *Candida albicans* and *Candida krusei* were found in the follow-up microbiological investigations (Table A2). Moreover, oral necrotic lesions were extensive during the next month and involved the left hemipalate (Figure 3).

The magnetic resonance examination of the head described pansinusitis, with complete obliteration of the left frontal, ethmoidal, and sphenoidal sinuses, thickening of mucous membrane of the maxillary sinuses, and an inhomogeneous fluid image of 12/24 mm, suggesting an abscess of the nasogenian region. The left eyeball was exophthalmic on the right side and there was infiltration of the internal right eye muscle, orbital, and infraorbital fat. In the left cortical and subcortical area was revealed an image of 10/17 mm, with ischemic character.

Due to pandemic COVID-19 restrictions, the hospital’s availability for a surgical procedure was limited. Moreover, the poor physical and mental condition of the patient and the general anesthesia high risk were considered, resulting in the decision to perform step-by-step small excisions of the necrotic tissues, adapted to the patient’s tolerance.

The tissue biopsy from the necrotic detritus of the left hemi-palate was obtained. The pathologic examination evidenced thick filamentous fungal structures, characterized by non-septate or rare transverse septa hyphae with irregular diameters and ramifications at right angles, with the appearance of mucormycosis. There were also noticed numerous associated microbial colonies, moderately diffuse inflammatory polymorphous infiltrates with lymphocytes, monocytes, and rare plasmacytes, comprising the stratified squamous mucosa. No dysplastic or neoplastic cells were observed.

The diagnosis supported by the clinic, histopathologic, and imaging examination was rhino-orbito-cerebral mucormycosis related to COVID-19.

The treatment strategy was complex: surgical interventions, antibiotics, antifungals, blood pressure control, nutritional and psychological support, and post-stroke rehabilitation. The antimicrobial medication contains of Piperacillin/Tazobactam (4 g/0.5 g tid) for 21 days aiming for the associated bacterial infections. The first-line option, amphotericin, was not available and oral treatment with Posaconazole began in the third month after COVID-19 (300 mg bid in the first day), for a six-month course (300 mg daily).

The surgical excisions continued step-by-step, until the complete removal of the necrotic tissues and safe margins was achieved, defined as exclusively healthy and normally vascularized tissues, following oncologic surgery principles. Consequently, the left hemi-palate tissues were lost, leading to a large oro-antral communication that was revealed in a computer tomography examination (Figure 4).

The external oxygen supplementation was required for two months post-COVID-19, the neurological disfunction was improved in six months, but the blindness of the left eye was irreversible. The biological markers are significantly improved (Table A1). After 9 months from COVID-19, the patient had a surgical reconstruction of the palate and dental prothesis, recovering mastication functionality and improving the quality of his life (Figure 5).

## 3. Discussion

Although the most mucorales infections are commonly distributed in the Asian region, the emergence of cases during the COVID-19 pandemic was reported in India, Iran, and other countries from the Americas and Europe. An early systematic review article on the diagnostic challenges of COVID-19 and management of mucormycosis from 2021, has analyzed 201 published cases, most of them (138 cases) from India, with 70 reported deaths [12].

A review on reports published up until April 2021, from 18 countries, analyzed 80 cases with CAM, most of them from India. All patients had a history of hospitalization for COVID-19 1–3 months ago, 79% experienced cortico-steroid treatment, and 49% were related to uncontrolled diabetes specifically with ketoacidosis. Rhino-orbital cerebral infection was developed in 74% cases, with mortality of 37%, which is lower than the 49% general mortality [13].

A meta-analysis reported in 2022 assessed 51 selected studies, including 2312 confirmed cases with of COVID-Associated Mucormycosis (CAM), with the median interval between COVID-19 and mucormycosis ranging from 10 to 31 days. The clinical main features identified diabetes co-morbidity (82%), rhino-orbital localization (90%) with orbital extension on 17% of patients, and an overall estimated mortality of 29% [14].

A sporadic mild increase of the glycemic values during the corticosteroid treatment were found in our patient, but diabetes was excluded. Meanwhile, diabetes is the main comorbidity in CAM, confirmed by most studies [15,16,17].

The mucormycosis and COVID-19 relationship could be explained by the virulence of the mucorales species and the ability and potency of the pathogenic mechanisms of the coronaviral infection and the mucoricin toxin. Experimentally, that proved to induce inflammation, vascular permeabilization, hypovolemic shock, and tissues necrosis. In vivo, the mucorales species increase the pro-inflammatory cytokines leading to the intensification of COVID-19 inflammation and the expression of specific virulence factors resulting from the host’s damaged tissues [4,18].

The altered host immune response in COVID-19 infection is also interesting for the specific antifungal defense, through lymphopenia and low response of T lymphocytes [18]. The non-specific immunity could be altered in severe disease by corticosteroid use, through the alteration in neutrophil migration, or ingestion and fusion of phagolysosomes [2,19,20].

The specific pathogenic mechanisms of COVID-19 result in endothelium damage of the vessels that are favorable for mucorales invasion [20]. The coronavirus could increase the expression of the co-receptor GRP78, involved either in the binding of viral spike protein to ACE-2 receptor, or the attachment of Rhizopus particles to the epithelial cells and invasion of the nasal epithelium [3,12].

Acidosis and hyperglycemia decrease phagocytosis and increase the release of free iron. When present, fungi have a high affinity to take up the free iron, and to use it as a growth factor [20,21]. An increased level of iron related to inflammation is regulated by hepcidin synthesized from the hepatocyte, and by intracellular sequestration and binding to transferrin, consecutive anemia, exacerbating hypoxia, and worsening of the COVID-19 disease [21].

Iatrogenic factors of CAM are corticosteroid therapy, biological therapies, CPAP, or dysbiosis after antibiotics [22].

Superinfections confirmed in our case, with *Kl. Pneumoniae*, *P. aeruginosa*, *C. albicans*, and *Candida krusei*, have a common feature known as the biofilm infection mechanism, which explains the failure of antibiotic treatment. *P. aeruginosa* and *C. albicans* are opportunistic pathogens associated in the mucosal tissue infections, and their coexistence is known to have existed for estimated billions of years. The interactions between these species evidenced by experimental studies could be synergistic or antagonistic, as well as quorum-sensing [23]. Virulence factors of *Pseudomonas* spp. alter the host epithelium in the intercellular tight junctions, while *Candida* spp. facilitates the growth of a *Pseudomonas* spp. biofilm [24]. Another interesting relationship was found between *P. aeruginosa* and *Rhizopus microsporus*, responsible for the most deaths in the disseminated infections. The potential control of bacteria on fungal growth is possible, since *P. aeruginosa* inhibits *R. microsporus* germination through sequestration of free environmental iron. On the contrary, the antibiotic treatment increases the risk for developing mucormycosis [25]. Polymicrobial infections increase the expression of proinflammatory cytokines and induce hyperinflammation status, with an unfavorable impact on morbidity and mortality. Host-related factors could influence the interaction between bacteria and fungi, but it is not clear how the immune response of the host influences the pathogenesis of polymicrobial infections [26].

The antifungal treatment was based on posaconazole, while it is commonly recommended as a second-line drug [5]. Posaconazole, similar to other triazole antifungal agents, acts by inhibition of the ergosterol synthesis pathway, but the activity against mucorales is variable, depending on the species. Posaconazole and isavuconazole exhibit higher activity against mucorales in vitro than other triazoles, but previous data demonstrated their limited clinical effectiveness [27]. Clinical pharmacodynamic studies for invasive mucormycosis are limited because of the rarity of the disease and the frequent lack of positive culture results. Mechanisms of acquired resistance remain largely unknown. Antifungal susceptibility testing of mucorales species notify a wide range of minimum inhibitory concentrations to the broad-spectrum azoles; therefore, the role and interpretation is not clarified enough [28].

The European Confederation of Medical Micology clinical registries reported successful first-line treatment with posaconazole in about 50–60% of patients with mucormycosis [5]. In a retrospective chart review performed in hospitalized patients from India with COVID-19-associated mucormycosis, a sub-therapeutic posaconazole level was observed in 24.1% of cases receiving this antifungal agent as a first-line drug. However, at 12 weeks follow-up, 72.4% patients had improved, 10.3% patients were stable, 10.3% patients died, and 6.9% patients left the hospital against medical advice, supporting posaconazole as first-line or alternate therapy of mucormycosis at sites with limited supplies of amphotericin B [29].

The present case is remarkable due to the epidemiological, clinical, diagnostic, and treatment particularities (Table 1).

## 4. Conclusions

Mucormycosis is a rare, severe disease, with progressive evolution and high mortality. The emergence of mucormycosis has advanced with the COVID-19 pandemic. The progression of mucormycosis depends on early diagnosis, promptly effective antifungal treatment, and surgical debridement. The management of mucormycosis calls for the collaboration of the multidisciplinary team: medical, surgical, imaging, and laboratory.

## Figures and Tables

**Figure 1 antibiotics-12-00031-f001:**
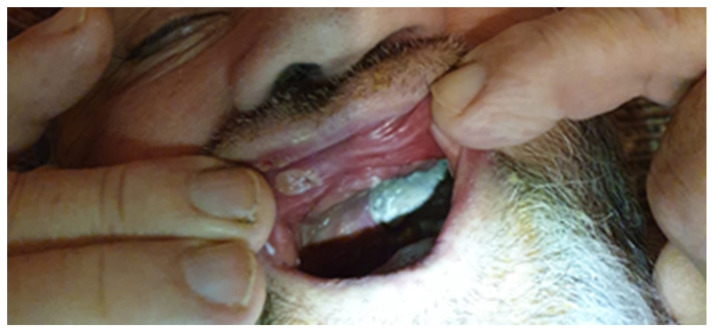
Light-gray mycotic plaques on the gums in a patient on discharge after COVID-19.

**Figure 2 antibiotics-12-00031-f002:**
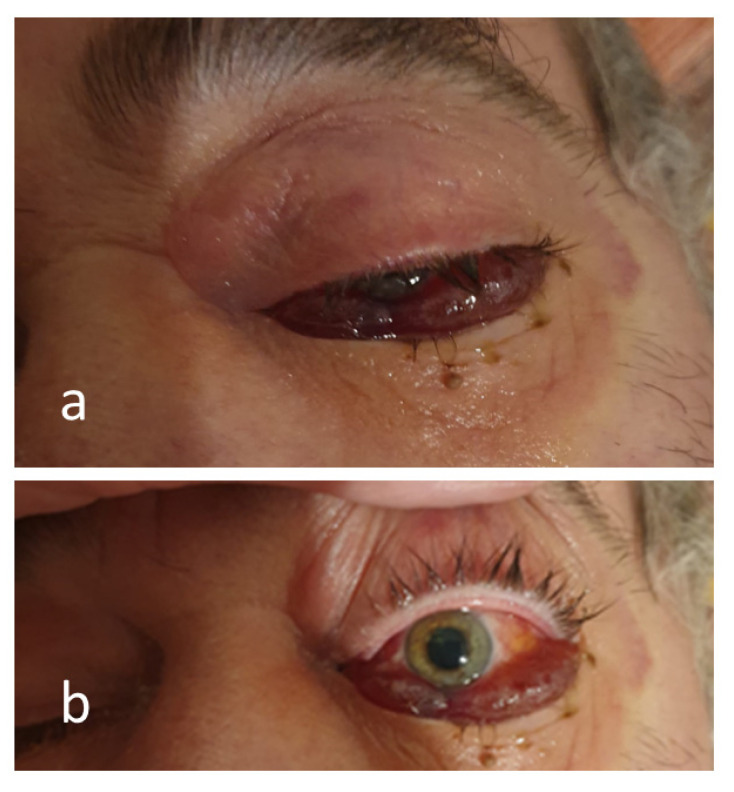
Images of the chemosis in the closed (**a**) and open (**b**) left eye, mydriasis (**b**) related to a left sided cerebrovascular accident, which occurred on the 28th day after COVID-19 diagnosis, complicated with mucormycosis.

**Figure 3 antibiotics-12-00031-f003:**
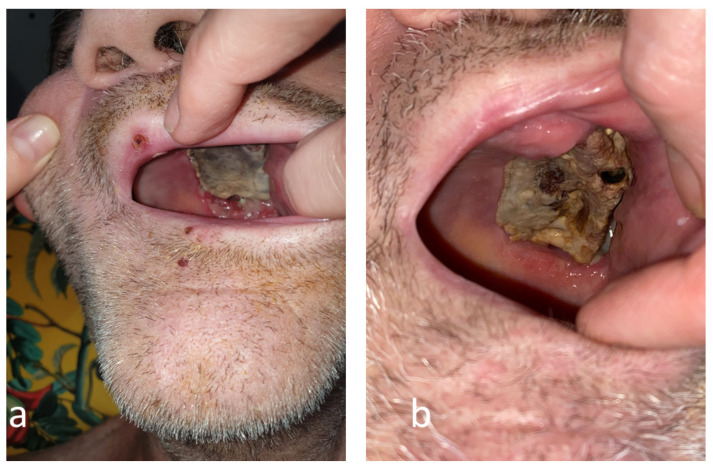
The evolution of the mucormycotic plaque in the second month (**a**) and third month (**b**) engaging the left hemi-palate.

**Figure 4 antibiotics-12-00031-f004:**
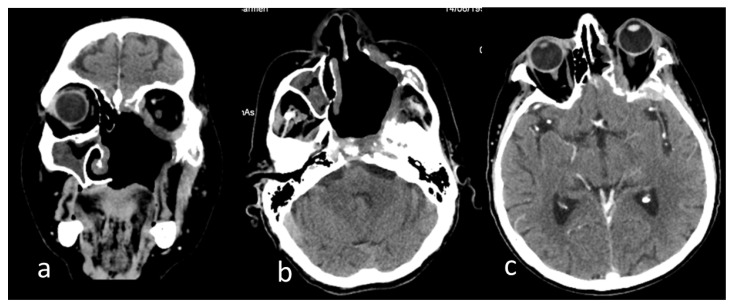
Computer tomography images of the head after the excision of the left hemi-palate mucormycotic plaque in native examination of the (**a**) coronal section, (**b**) axial section, and (**c**) postcontrast examination axial section. Dysmorphic appearance with bone defects at the maxillary, ethmoidal, sphenoidal, temporal, and zygomatic levels on the left side (**a**,**b**); patchy thickening of the mucosa of the right maxillary, ethmoidal sinuses (**a**,**b**); fibrotic densification in the left temporal fossa and left exophthalmos (**b**,**c**); left temporal thickening and small hypodense area in the neighboring parenchyma (**c**).

**Figure 5 antibiotics-12-00031-f005:**
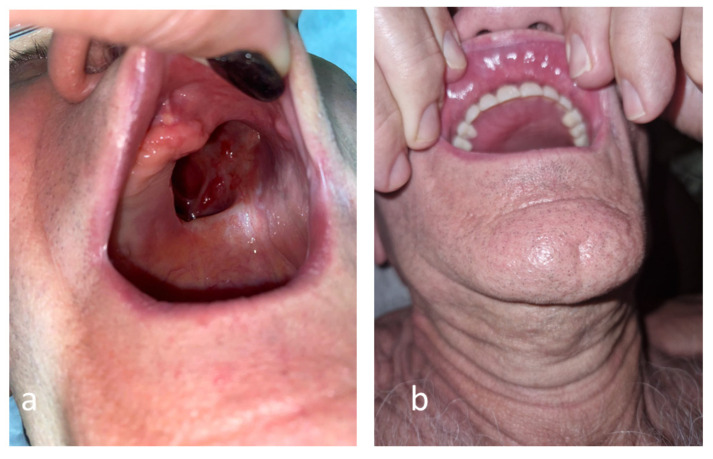
Anatomical defect of the palate related to rhino-orbital mucormycosis before (**a**) and after (**b**) the surgical reconstruction.

**Table 1 antibiotics-12-00031-t001:** Particularities of the reported case.

Mucormycosis occurred in a geographical region where this fungal infection is unusual. The patient was an old man but had no diabetes or other immunosuppression conditions. Rhino-orbital localization is the common presentation of CAM. Co-infections with Pseudomonas aeruginosa, Klebsiella pneumoniae and *Candida* spp. are associated with biofilms and have complicated the management of the patient. The delayed diagnostic due to the difficulty recognizing lesions have been associated with neurologic complications, anatomical defects, and oral dysfunctions. The antifungal treatment was based on posaconazole with afavorable result, although is commonly recommended as a second-line drug. The duration of antifungal therapy was 6 months, considering the resolution of signs, symptoms, and paraclinical improvement, but the guidelines do not define how long the antifungal treatment should be continued. The fungal infection was cured with sequelae of bone and soft tissues loss, requiring extensive and difficult reconstruction interventions that were successfully achieved.

## Data Availability

Data is contained within the article.

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
