# Peer review of "Diagnostic and Treatment Challenges of Emergent COVID-Associated-Mucormycosis: A Case Report and Review of the Literature"

_antibiotics, 2022, doi:10.3390/antibiotics12010031_

Round 1

Reviewer 1 Report

The authors present a case of mucormycosis in a patient with recent COVID-19 infection and review the literature. There is no doubt that it is a very illustrative and well-documented case. One of the peculiarities they point out is the antifungal treatment with oral posaconazole due to the unavailability of liposomal amphotericin b, and they indicate that this treatment is considered second-line. It would be very interesting if the authors could carry out a literature review of previous cases in which posaconazole was used as initial treatment and the results obtained. Similarly, the manuscript could also be enriched by analysing the current positioning of azoles such as posaconazole and isavuconazole in the most recent guidelines published on the treatment of mucormycosis.

Author Response

Response to Reviewer 1 Comments

According to the comments and suggestions, we have included the supplimentary data to “Discussion” section:

Point 1: It would be very interesting if the authors could carry out a literature review of previous cases in which posaconazole was used as initial treatment and the results obtained.

Response 1:

The European Confederation of Medical Micology clinical registries reported successful first-line treatment with posaconazole in about 50-60% of patients [Cornely OA].

A retrospective chart review performed in hospitalised patients from India, supports the  posaconazole as first-line or alternate therapy to treat mucormycosis during limited supply of amphotericin B. From 29 cases with COVID-19 associated Mucormycosis, at 12 weeks follow-up, 72.4% patients were improved, 10.3% patients were stable, 10.3% patients died and 6.9% patients left hospital against medical advice, while .sub-therapeutic posaconazole through level was observed in 24.1% cases.[Patel, 2022]

Cornely OA, Alastruey-Izquierdo A, Arenz D, Chen SCA, Dannaoui E, Hochhegger B, et al. Global guideline for the diagnosis and management of mucormycosis: an initiative of the European Confederation of Medical Mycology in cooperation with the Mycoses Study Group Education and Research Consortium. Lancet Infect Dis. 2019;19(12):e405-e421. doi: 10.1016/S1473-3099(19)30312-3.

Patel A, Patel K, Patel K, Shah K, Chakrabarti A. Therapeutic drug monitoring of posaconazole delayed release tablet while managing COVID-19-associated mucormycosis in a real-life setting. Mycoses. 2022;65(3):312-316. doi: 10.1111/myc.13420.

Point 2: Similarly, the manuscript could also be enriched by analysing the current positioning of azoles such as posaconazole and isavuconazole in the most recent guidelines published on the treatment of mucormycosis.

Response 2:

The antifungal treatment was based on posaconazole, while it is commonly recommended as a 2nd-line drug [7]. Posaconazole, as other triazoles antifungal agents, is acting by inhibition of the ergosterol synthesis pathway, but the activity against Mucorales is variable, depending on the species. Posaconazole and isavuconazole exhibit higher activity against Mucorales in vitro than other triazoles, but previous data demonstrated the limited clinical effectiveness [Smith, 2022]. Clinical pharmacodynamic studies for invasive mucormycosis are limited because of the rarity of the disease and the frequent lack of positive culture results. Mechanisms of acquired resistance remain largely unknown. Antifungal susceptibility testing of Mucorales species notify a wide range of minimum inhibitory concentration to the broad-spectrum azoles, therefore the role and interpretation is not enough clarified [Laroth, 2021].

Cornely OA, Alastruey-Izquierdo A, Arenz D, Chen SCA, Dannaoui E, Hochhegger B, et al. Global guideline for the diagnosis and management of mucormycosis: an initiative of the European Confederation of Medical Mycology in cooperation with the Mycoses Study Group Education and Research Consortium. Lancet Infect Dis. 2019;19(12):e405-e421. doi: 10.1016/S1473-3099(19)30312-3.

Smith C, Lee SC. Current treatments against mucormycosis and future directions. PLoS Pathog. 2022 Oct 13;18(10):e1010858. doi: 10.1371/journal.ppat.1010858. 

Lamoth F, Lewis RE, Kontoyiannis DP. Role and Interpretation of Antifungal Susceptibility Testing for the Management of Invasive Fungal Infections. Journal of Fungi. 2021; 7(1):17. https://doi.org/10.3390/jof7010017

Reviewer 2 Report

The manuscript titled: Diagnostic and Treatment Challenges of Emergent Covid-Associated-Mucormicosis: A case report and review of the literature is an interesting work where the relationship between covid and mucormycosis is analyzed regarding a case in an immunocompetent patient without diabetes. Several unusual details of the patient's presentation and management make the case even more interesting, such as associated infections, posaconazole antifungal treatment, and patient survival. I congratulate the authors for this success story.

My main criticism is that the etiological agent was not identified. It would be interesting to know, since this is an immunocompetent person without diabetes.

Minor comments

Page 2 line 45  change identified species to identified genera.

Page 3 line 106 change Clostridioides to Clostridium.

Page 4 line 129 change krusey to krusei.

Page 7 lines 196-199 It seems that these lines are not part of the discussion, please review them.

Please make sure that the names of the microorganisms appear in italics throughout the text.

The words spp. or sp. should not be in italics, please check throughout the document.

References must be in the journal format.

The names of microorganisms are written in full the first time (not counting the abstract), then they are abbreviated. For example Pseudomonas aeruginosa the first time and then P. aeruginosa. Please review this throughout the document.

Author Response

Response to Reviewer 2 Comments

Minor comments

Point 1: Page 2 line 45  change identified species to identified genera.

Response 1: Correction done.

Point 2: Page 3 line 106 change Clostridioides to Clostridium.

Response2: Correction done.

Point 3: Page 4 line 129 change krusey to krusei.

Response 3 : Correction done.

Point 4: Page 7 lines 196-199 It seems that these lines are not part of the discussion, please review them.
Respondse 4: Deleated paragraph.

Point 5: Please make sure that the names of the microorganisms appear in italics throughout the text.

The words spp. or sp. should not be in italics, please check throughout the document.

Response 5: We revised

Point 6: References must be in the journal format.

Response 6: We revised

Point 7: The names of microorganisms are written in full the first time (not counting the abstract), then they are abbreviated. For example Pseudomonas aeruginosa the first time and then P. aeruginosa. Please review this throughout the document.

Response 7: We revised.

Reviewer 3 Report

This manuscript reported a case with mucormicosis in COVID-19 epidemic era. Some comments appeared below.

1. All scientific name should be italics throughout the manuscript.

2. All paragraphs should be reorganized for their appropriate in the Introduction section.

3. Line 46-50 and 71-75 should be combined.

4. Introduction: How many cases of mucormicosis up to the present? which geographic regions have high incidence? How many cases per year?

5. What is route of infection of this case?

6. Discussion: Line 196-199 should be removed.

Author Response

Response to Reviewer 3 Comments

Point 1. All scientific name should be italics throughout the manuscript.

Response 1: We revised.

Point 2. All paragraphs should be reorganized for their appropriate in the Introduction section.

Response 2: We added supplimentary data on regional incidence.

Point 3. Line 46-50 and 71-75 should be combined.

Response 3: We combined the paragraphs.

  1. Introduction: How many cases of mucormicosis up to the present? which geographic regions have high incidence? How many cases per year?

Response 3: Supplimentary data (introduction section):

The real burden of human mucormycosis is unknown, because is not a reportable disease  [Prakash, 2019]. A review of 851 cases colected from January 2000 to January 2017 evidenced the majority of the cases from Europe(34%), followed by Asia (31%) and North or South America (28%) [Jeong, 2019]. However, the disease is more frequent  in the  developing countries, but the diagnostic is underestimated in this region, mainly due to laboratory sub-optimal facility [Prakash, 2019].

A report from 2016 estimated an annual incidence of mucormycosis in Romania of 0.04/100000 [Mares, 2018]. The global prevalence of mucormycosis estimates of year 2019-2020 range between 0.005 and 1.7 per million population, wehereas the highest prevalence is noticed in India (0.14 per 1000) [Singh, 2021]. Growing number of mucormycosis reported cases during the COVID-19 pandemic in creased the interest of the public health.

Rferences:

Prakash H, Chakrabarti A. Global Epidemiology of Mucormycosis. J Fungi (Basel). 2019 Mar 21;5(1):26. doi: 10.3390/jof5010026. 

Jeong W., Keighley C., Wolfe R., Lee W.L., Slavin M.A., Kong D.C.M., Chen S.C.A. The epidemiology and clinical manifestations of mucormycosis: A systematic review and meta-analysis of case reports. Clin. Microbiol. Infect. 2019;25:26–34. doi: 10.1016/j.cmi.2018.07.011.

MareÈ™ M, Moroti-Constantinescu VR, Denning DW. The Burden of Fungal Diseases in Romania. J Fungi (Basel). 2018 Mar 1;4(1):31. doi: 10.3390/jof4010031.

Singh AK, Singh R, Joshi SR, Misra A. Mucormycosis in COVID-19: A systematic review of cases reported worldwide and in India. Diabetes Metab Syndr. 2021;15(4):102146. doi: 10.1016/j.dsx.2021.05.019.

  1. What is route of infection of this case?

Response 5: The route of infection was not identified by the epidemiological investigation, but is probably related to the procedures of respiratory support during COVID-19.

  1. Discussion: Line 196-199 should be removed.

Response 6: We deleted the paragraph.
